# Oral processing behavior and dental caries; an insight into a new relationship

**Melanie F. Alazzam**[1]*, **Issam B. Rasheed**[1], **Suhad H. Aljundi**[2], **Dalal A. Shamiyah**[3], **Yousef S. Khader**[4], **Reem S. Abdelhafez**[2], **Mohammad S. Alrashdan**[1,5]

1 Department of Oral Medicine and Oral Surgery, Jordan University of Science and Technology, Irbid, Jordan, 2 Department of Preventive Dentistry, Jordan University of Science and Technology, Irbid, Jordan, 3 Undergraduate Bachelor of Dental Surgery Program, Jordan University of Science and Technology, Irbid, Jordan, 4 Department of Public Health, Jordan University of Science and Technology, Irbid, Jordan, 5 Department of Oral and Craniofacial Health Sciences, University of Sharjah, Sharjah, United Arab Emirates

* mfalazam@just.edu.jo

**Data Availability Statement:** All relevant data are within the manuscript and its Supporting Information files.

## Abstract

### Introduction

Previous evidence suggests an individual variation in the preferred oral processing behavior. Individuals can be classified as firm processing(FPL) or soft processing likers(SPL). FPL (crunchers and chewers) prefer using their teeth while SPL(smooshers and suckers) prefer using the tongue and the palate when processing different food items. Variation in the preferred oral processing behavior has been associated with differences in food texture preference and eating time. Time is one of the factors directly related to the development of dental caries(tooth decay). Oral retention and eating times are associated with greater caries experience. This study aims to explore if a relationship exists between the preferred oral processing behavior and the individual's caries experience.

### Materials and methods

This was a cross-sectional, dental center-based study conducted at Jordan University of Science and Technology. Five hundred participants consented to fill out the preferred oral processing behavior(POPB) questionnaire. Anthropometric measurements (including weight, height, and waist circumference) were recorded. A single trained and calibrated dentist registered each participant's caries experience and plaque levels using the DMFS index and plaque index of Silness and Loe.

### Results

A total of 351(70.2%) and 149(29.8%) participants were typed as FPL and SPL, respectively. SPL demonstrated higher levels of dental caries experience compared to FPL. The mean DMFS score for SPL was 28.8(±25.43) while for FPL was 18.71(± 18.34). This difference remained significant after adjustment for confounders(P<0.001). SPL exhibited a significantly higher mean score for the "M" component(P <0.001) while no significant difference in the mean score of the "D"(P = 0.076) and "F"(P = 0.272) components was observed when compared to FPL.

**Funding:** This research was supported by Jordan University of Science and Technology (JUST), (NFRG # 20210057), awarded to MFA, and was documented at the deanship of research (JUST). The funders had no role in study design, data collection and analysis, decision to publish, or preparation of the manuscript.

**Competing interests:** The authors have declared that no competing interests exist.

## Conclusion

The current findings provide new insight into a possible relationship between the preferred oral processing behavior and an individual's caries experience. A relationship in which the preferred oral processing behavior can potentially affect and/or be affected by the dental caries experience.

## Introduction

Human beings engage in food oral processing everyday [1]. It is not only important for food intake and digestion but also for the sensory experience associated with food intake (i.e. perception of texture and flavor) [2–5]. Food oral processing is defined as a series of complex processes taking place in the oral cavity. The processes lead to the formation of a bolus suitable for swallowing. During food oral processing, the food is manipulated by the tongue and the teeth and is continuously mixed with saliva [1, 2]. Both instrumental [6] and introspective [7–9] measurements suggest an individual variation in oral processing and manipulation [3, 9, 10].

Using a questionnaire and visual aids (Jeltema/Beckley Mouth Behavior tool; JBMB® tool), four mouth behavior (oral processing) groups were identified [7, 8, 11]. Based on the tool, individuals were classified as crunchers, chewers, smooshers or suckers. These groups were broadly categorized by Cattaneo et al. into two main oral processing behaviors; firm processing likers (FPL) and soft processing likers (SPL) [12, 13]. FPL (crunchers and chewers) preferred using their teeth when breaking down the food [12]. Crunchers were more forceful in their biting and preferred crunchy food (food that fractures upon biting). Chewers preferred food that could be chewed and didn't fracture on biting [8, 12]. SPL (smooshers and suckers) preferred to manipulate (process) the food between the tongue and the palate and didn't prefer using their teeth [8, 12]. Smooshers preferred food items (such as puddings, creamy candies) that spread all over the oral cavity and were kept for a long time in the mouth. Suckers preferred hard food (hard candies) that could be placed between the tongue and the palate allowing the food item to be sucked on for a long time.

As mouth behavior groups varied in the preferred way of processing food items, they demonstrated differences in food texture preference [8, 11, 14] and the eating time [8, 15]. Qualitative and quantitative research indicated SPL were slow eaters [8, 15]. Indeed, SPL needed a longer time to eat chocolate compared to FPL. The eating time was significantly greater among SPL [15]. Keeping food for a long time in the mouth was a feature characterizing SPL [8]. When food is retained for a longer duration, individuals are at greater risk of developing dental caries (tooth decay) [16–19].

Dental caries is a chronic non-communicable disease affecting the crown and/or the root of deciduous and permanent teeth [20, 21]. It is the most prevalent health condition worldwide affecting around 2.4 billion individuals [22–24]. Untreated carious lesions may lead to dental pain, pulpal/periapical diseases and tooth loss [25–28]. These problems can be avoided by proper management and prevention of dental caries. Such steps require developing a better understanding of the dental caries process and its risk factors [20, 29].

Tooth decay is defined as a localized destruction of the mineralized dental hard tissue (enamel, dentine, cementum) by weak organic acids produced in the mouth [30, 31]. The acids are generated by biofilm bacteria during the metabolism of dietary fermentable carbohydrates. A process that will cause a drop in the PH below the critical level leading to the demineralization of the tooth surface [30, 31]. An imbalance in the demineralization-

remineralization process (minerals available in the saliva) will lead to the development and progression of carious lesions [30, 31].

The etiology of dental caries is complex and multifactorial in nature [31, 32]. Dental caries results from the interplay over time between cariogenic biofilm bacteria that produce acids, substrate (fermentable carbohydrates), the tooth structure, and saliva [30, 33, 34]. This interaction is modified by different risk and protective factors including psychological, social, environmental, behavioral, biological, and genetic influences [20, 31, 35, 36].

Time is one of the etiological factors directly contributing to the etiology of dental caries [30, 33, 36]. The cariogenicity of ingested food (sugars) increases with prolonged oral retention time [17, 18, 37]. As food is kept in the mouth for a longer period, teeth will be exposed to a longer duration of acid production, demineralization, and a shortened duration of remineralization [17, 38]. Evidence suggests that a longer oral retention time of fermentable carbohydrates (sugar, starch) is more likely to produce dental carious lesions [16, 18]. Moreover, longer eating time is associated with greater caries experience [39, 40].

A longer eating time was one of the features characterizing SPL [8, 15]. Therefore, this study aimed to investigate if a relationship exists between the preferred oral processing behavior (FPL vs SPL) and dental caries by examining caries experience among these groups. Although the study does not aim to establish a cause-effect relationship, it aims to shed light on a potential link between the oral processing behavior and dental caries. To the best of our knowledge, no other study examined such a relationship. Our null hypothesis states that there is no difference in caries experience among the different oral processing behaviors.

## Materials and methods

### Study design and sample

This cross-sectional study was conducted in two dental teaching centers at Jordan University of Science and Technology (JUST), Irbid, Jordan. A consecutive sample of 500 random participants aged 18–64 years old were enrolled between November 14th 2021 and August 10th 2022. Individuals with any of the following criteria were excluded: being allergic to certain foods, on diet restriction (Vegans, Keto, etc.), wearing a removable/fixed orthodontic appliance, wearing a partial/complete denture, and being non-Jordanian. After explaining the aims and the nature of the study, written informed consent was obtained.

Upon enrollment, each participant answered two questionnaires: the general information questionnaire and the preferred oral processing behavior (POPB) questionnaire. In addition, anthropometric measurements, dental caries, and plaque levels were recorded.

Assuming a pooled standard deviation of 20 for DMFS, the study would require a sample size of 63 for each group to achieve a power of 80% and a level of significance of 5% (two-sided), for detecting a true difference of 10 units in DMFS between the study groups. The sample size increased significantly to adjust for multiple covariates in the General Linear Model. The number of recruited subjects is much higher than the required sample size for cluster analysis.

The study protocol was approved by the Institutional Review Board at JUST (IRB # 18/137/2021, Date 14.1.2021). It was conducted in accordance with the World Medical Declaration of Helsinki and conformed to the STROBE statement for observational studies.

### Study questionnaires

All questionnaires were developed using Google Forms. They were administered anonymously in the presence of a trained research assistant. The assistants checked that participants understood and answered all questions.

The general information questionnaire inquired about the participant's general demographics, medical problems and medications, and oral health behaviors. The oral health behavior was examined from different aspects. The first two aspects included inquiries about participant's oral hygiene habits (brushing and flossing frequency) and the frequency of dental visits. The third aspect was assessing the nutritional preferences of study participants. The nutritional preferences were assessed by examining the daily/weekly intake frequency of sugary drinks, candies, snacks, and sugary baked items.

The preferred oral processing behavior (POPB) questionnaire was adapted from Cattaneo et al. [12]. The questionnaire was composed of 20 text and 4 photo-based questions. The questions addressed the different behaviors that individuals prefer to use when manipulating food in their mouths. Study participants expressed their agreement/disagreement with each statement using a 6-point Likert scale anchored with the labels "strongly agree" (1) to "strongly disagree" (6). A key was used to further elaborate on the meaning of each label on the scale (S1 File). The questionnaire was translated into Arabic language by a committee of a bilingual linguist and 2 other dental academics. The Arabic-translated version of the questionnaire was distributed among a pilot sample of 23 participants. Based on the pilot testing and the experts' opinion, two text questions and three of the photo-based questions were slightly modified to include food items that were more familiar to our study sample (changes highlighted in supplementary material S2 File).

## Anthropometric measurements

Anthropometric measurements were recorded by a trained research assistant. The measurements included the height, weight, and waist circumference of each participant while being barefoot and wearing light clothing. The weight was measured in kilograms (kg) using an electronic scale (Tanita ® BC-730, Tanita Corporation, Tokyo, Japan) to the nearest 0.1 kg. The height was measured in centimeters (cm) using a height rod (Tanita ® HR-001, Tanita Corporation, Tokyo, Japan) to the nearest 0.1 cm. Waist circumference was recorded by using a non-stretching measuring tape placed at a level midway the distance between the lowest rib margin and the iliac crest to the nearest 0.5 cm [41, 42]. Body Mass Index (BMI) was calculated by dividing the weight in kg by the height in meters squared [43].

## Dental plaque and caries assessment

Caries experience was assessed using the Decayed-Missing-Filled-Surface (DMFS) index [44] following the WHO guidelines [45]. Five surfaces of molars and premolars and 4 surfaces of incisors and canines were examined (excluding third molars). The DMFS score values ranged from 0–128 (where 128 was the highest caries experience). Participants were examined in the dental clinic by a trained and calibrated licensed dentist. No radiographs were taken.

The amount of dental plaque was assessed in all teeth using the plaque index of Silness and Loe [46–48]. The score was recorded by a trained and calibrated licensed dentist for 4 surfaces of each tooth: buccal, lingual, mesial, and distal surface [46, 49, 50]. The mean plaque index score for each tooth was calculated by dividing the total scores for all examined surfaces by 4 [46, 49, 50]. The individual's plaque index was calculated by adding the plaque indices of all teeth divided by the total number of the examined teeth [46, 49, 50].

## Training and calibration

For dental caries registration, the observer was trained by a pediatric specialist in two sessions. The first session included a lecture that focused on caries registration using the DMFS index. The second session involved a training exercise on caries assessment (the DMFS index) using

clinical pictures. The specialist and the observer examined a group of volunteer patients at JUST dental clinics (n = 20) before starting data collection. Using the intra-class correlation, the inter-rater reliability was calculated at 0.98 and the intra-rater reliability was calculated at 0.99.

For the assessment of dental plaque, the same observer had a training session with a peri-odontist regarding the plaque index of Silness and Loe. The session included a thorough explanation of the plaque index and clinical training on a volunteer patient at JUST dental clinics. For calibration purposes, the observer assessed the plaque levels of Ramfjord teeth in a group of volunteer patients (n = 20). The intra-rater reliability (ICC = 0.89) was good.

## Statistical analysis

Data analysis was carried out using IBM SPSS Statistics, version 24. To achieve the segmentation of the study subjects based on their preferred oral processing behavior, we employed K-Means clustering due to its well-documented simplicity and efficiency. The optimal number of clusters (k) was determined through the application of the elbow method, a commonly accepted technique for finding a balance between intra-cluster similarity and inter-cluster dissimilarity. After a thorough evaluation of different k values, it was determined that k = 2 yielded the most interpretable results. This resulted in the identification of two distinct groups, namely "Firm Processing Likers" and "Soft Processing Likers." The assignment of subjects to these clusters was accomplished by applying the K-Means algorithm to the standardized data, with each subject being allocated to one of the two clusters based on their proximity to the cluster centroids. To assess the robustness of our cluster solution, a sensitivity analysis was conducted. This involved varying the number of clusters (k) to ensure the consistency and stability of the key patterns within the data. Seventeen questions of the POPB questionnaire helped identify each cluster (Q2, Q3, Q4, Q5, Q8, Q10, Q11, Q12, Q14, Q15, Q17, Q19, Q20, Q21, Q22, Q23, Q24).

Categorical variables were compared across subjects, categorized by their preferred oral processing behavior, using the Chi-square test. Furthermore, the General Linear Model (GLM) was employed to evaluate differences in continuous outcome variables between the two subject groups while adjusting for important covariates. The GLM procedure provides regression analysis and analysis of variance for one dependent variable by one or more factors and/or variables. A significance level of $p < 0.05$ was considered to indicate statistical significance.

## Results

### Participants' assignment to the preferred oral processing behavior

Based on the preferred oral processing behavior questionnaire, the cluster analysis identified 2 groups: SPL and FPL. A total of 149 (29.8%) participants were SPL and 351 (70.2%) participants were FPL.

### Characteristics of the oral processing behavior groups

Five hundred participants with an average age of 31.2 years completed the study. Males comprised 42.4% (n = 212) and females comprised 57.6% (n = 288) of the study sample. Of males, 72.2% (n = 153) were FPL and 27.8% (n = 59) were SPL. Among females, 68.8% (n = 198) were FPL and 31.3% (n = 90) were SPL (Table 1). There was no statistically significant difference between males and females in their preferred oral processing behavior (P = 0.409).

Similarly, there was no statistically significant difference between preferred oral processing behavior and the participant's age group (P = 0.053) (Table 1).

**Table 1. General characteristics of the oral processing behavior groups.**

| Attribute | | Oral Processing Behavior Group | | P-Value |
|---|---|---|---|---|
| | | Firm Processing Likers | Soft Processing Likers | |
| | | n (%) | n (%) | |
| **Sex** | Male | 153 (72.2%) | 59 (27.8%) | 0.409 |
| | Female | 198 (68.8%) | 90 (31.3%) | |
| **Age Group** | <20 | 66 (79.5%) | 17 (20.5%) | 0.053 |
| | 20–29 | 146 (73%) | 54 (27%) | |
| | 30–39 | 46 (63.9%) | 26 (36.1%) | |
| | 40–49 | 47 (60.3%) | 31 (39.7%) | |
| | 50+ | 46 (68.7%) | 21 (31.3%) | |
| **BMI** | Normal | 157 (74.1%) | 55 (25.9%) | 0.185 |
| | Overweight | 97 (65.1%) | 52 (34.9%) | |
| | Obesity | 97 (69.8%) | 42 (30.2%) | |
| **Medical Problems** | Yes | 38 (61.3%) | 24 (38.7%) | 0.101 |
| | No | 313 (71.5%) | 125 (28.5%) | |
| **Diabetes** | Yes | 13 (54.2%) | 11 (45.8%) | 0.078 |
| | No | 338 (71%) | 138 (29%) | |
| **Hypertension** | Yes | 24 (63.2%) | 14 (36.8%) | 0.323 |
| | No | 327 (70.8%) | 135 (29.2%) | |
| **Participant's Education Level** | Less than Basic- Intermediate | 125 (71.4%) | 50 (28.6%) | 0.659 |
| | Advanced (Ex. Diploma, Bachelor's, Master's, Doctoral levels) | 226 (69.5%) | 99 (30.5%) | |
| **Income Level** | Irregular income | 33 (66%) | 17 (34%) | 0.806 |
| | <500 JD | 137 (70.6%) | 57 (29.4%) | |
| | ≥500 - <1000 JD | 128 (71.9%) | 50 (28.1%) | |
| | ≥1000 - <2000 JD | 33 (64.7%) | 18 (35.3%) | |
| | ≥2000 JD | 20 (74.1%) | 7 (25.9%) | |
| **Smoking Status** | Yes | 116 (73%) | 43 (27%) | 0.358 |
| | No | 235 (68.9%) | 106 (31.1%) | |

The general health of the different oral processing behavior groups was investigated. There was no significant difference in the existence (P = 0.101) and type of medical problems reported by both groups; hypertension (P = 0.323) and diabetes (P = 0.078) (Table 1). Furthermore, there was no significant difference between FPL and SPL in overall obesity levels, as measured by BMI (Table 1). The proportion of normal, overweight, and obese participants among different oral processing behavior groups was not significantly different (P = 0.185).

Regarding socioeconomic status (SES), there was no significant difference in education (P = 0.659) and income levels (P = 0.806) among the different oral processing behavior groups (Table 1).

Smoking as a social habit was investigated. Smokers comprised 73% of FPL and 27% of SPL. Non-smokers comprised 68.9% of FPL and 31.1% of SPL. There was no significant difference in the proportion of smokers among FPL and SPL (P = 0.358) (Table 1).

## Oral Health behavior of the oral processing groups

The oral health behavior was not significantly different among the preferred oral processing behavior groups (Table 2). SPL and FPL did not differ significantly in brushing (P = 0.605) and flossing frequencies (P = 0.356). Moreover, both groups demonstrated similar routine dental attendance (P = 0.323). Concerning nutritional preferences, no significant difference in

**Table 2. Oral health behavior of the oral processing behavior groups.**

| Oral Health Behavior | | Oral Processing Behavior Group | | P-Value |
|---|---|---|---|---|
| | | **Firm Processing Likers** | **Soft Processing Likers** | |
| | | **n (%)** | **n (%)** | |
| **Brushing Frequency** | Doesn't brush | 30 (8.5%) | 14 (9.4%) | 0.605 |
| | 1–2 times/week | 48 (13.7%) | 17 (11.4%) | |
| | 3–5 times/week | 38 (10.8%) | 15 (10.1%) | |
| | 1–2 times/ day | 206 (58.7%) | 85 (57%) | |
| | 3 times/ day | 26 (7.4%) | 14 (9.4%) | |
| | After every meal | 3 (0.9%) | 4 (2.7%) | |
| **Flossing Frequency** | Doesn't floss | 268 (76.4%) | 103 (69.1%) | 0.356 |
| | 1–2 times/week | 36 (10.3%) | 20 (13.4%) | |
| | 3–5 times/week | 14 (4%) | 4 (2.7%) | |
| | 1–2 times/ day | 20 (5.7%) | 15 (10.1%) | |
| | 3 times/ day | 1 (0.3%) | 1 (0.7%) | |
| | After every meal | 12 (3.4%) | 6 (4%) | |
| **Dental Visits** | Never | 81 (23.1%) | 22 (14.8%) | 0.323 |
| | 1 time or less/ year | 123 (35%) | 57 (38.3%) | |
| | 1 time/ year | 43 (12.3%) | 21 (14.1%) | |
| | 2 times/ year | 51 (14.5%) | 26 (17.4%) | |
| | 3 times or more/year | 53 (15.1%) | 23 (15.4%) | |
| **Consumption Frequency of Sugary Drinks** | Never | 38 (10.8%) | 23 (15.4%) | 0.165 |
| | 1 time or less/week | 130 (37%) | 45 (30.2%) | |
| | 2–3 times/ week | 81 (23.1%) | 43 (28.9%) | |
| | More than 3 times/week | 102 (29.1%) | 38 (25.5%) | |
| **Consumption Frequency of Salty Snacks** | Never | 49 (14%) | 32 (21.5%) | *0.010 |
| | 1 time or less/week | 117 (33.3%) | 59 (39.6%) | |
| | 2–3 times/ week | 91 (25.9%) | 36 (24.2%) | |
| | More than 3 times/week | 94 (26.8%) | 22 (14.8%) | |
| **Consumption Frequency of Candies** | Never | 33 (9.4%) | 13 (8.7%) | 0.102 |
| | 1 time or less/week | 95 (27.1%) | 52 (34.9%) | |
| | 2–3 times/ week | 81 (23.1%) | 40 (26.8%) | |
| | More than 3 times/week | 142 (40.5%) | 44 (29.5%) | |
| **Consumption Frequency of Sweet-Baked Items** | Never | 34 (9.7%) | 19 (12.8%) | 0.391 |
| | 1 time or less/week | 167 (47.6%) | 78 (52.3%) | |
| | 2–3 times/ week | 86 (24.5%) | 30 (20.1%) | |
| | More than 3 times/week | 64 (18.2%) | 22 (14.8%) | |

the consumption frequency of sugary drinks (P = 0.165), candies (P = 0.102), and sugary baked (P = 0.391) items was found between the groups. However, FPL consumed salty snacks (potato chips) more frequently than SPL (P = 0.010).

## Dental plaque and caries experience among the oral processing behavior groups

The mean (± SD) caries experience in our sample as scored by the DMFS index was 21.72 (± 21.19). In the GLM analysis, SPL demonstrated higher levels of caries experience compared to FPL (P<0.001) after adjusting for age, sex, BMI, medical problems, medications, dental visits, brushing and flossing frequency, consumption frequency of sugary drinks, candies, snacks,

and sugary baked item, smoking status, participant's education level, mothers' education level, income level, and the plaque index. The mean DMFS score for SPL was 28.80 (± 25.43) while that for FPL was 18.71 (± 18.34) (Table 3).

Although caries experience was different between the oral processing groups, dental plaque levels did not differ significantly after adjusting for other variables (P = 0.964). The mean plaque score in SPL was identical to that of FPL 0.61(± 0.60). Plaque levels did not differ significantly among different oral processing groups when assessing them in the maxillary (P = 0.978) and mandibular (P = 0.968) arches, maxillary right (P = 0.910) and left (P = 0.836) quadrants, mandibular right (P = 0.902) and left (P = 0.986) quadrants (Table 3).

## Discussion

The current study was conducted to achieve two aims. The first was to explore the preferred oral processing behavior among Jordanians. The second aim was to investigate whether there is a relationship between an individual's preferred oral processing behavior and their dental caries experience. The study did not intend to explore a cause-effect association but rather to give a new insight into a potential factor that may affect or be affected by dental caries. This work has demonstrated three key findings. The first key finding indicated that a greater percentage of Jordanians (70.2%) were classified as FPL. This was similar to other populations; Danish [12] and North American populations [8, 51, 52]. On the contrary, Jordanians were different from a Chinese population in which the majority (77%) were SPL [12]. This observed variation could be attributed to differences in cultural and dietary habits associated with food consumption [12, 53] and/or to the dental status of the study population [54–57].

The second key finding was that the preferred oral processing behavior was not influenced by an individual's age or sex. This finding was previously reported in several studies. For

**Table 3. Multivariate analysis of caries experience and plaque levels among oral processing behavior groups using the General Linear Model (GLM) procedure.**

| Type of Assessment | Oral Processing Behavior Group | | P-Value |
|---|---|---|---|
| **Caries Assessment** | **Firm Processing Likers** | **Soft Processing Likers** | |
| **(DMFS index)** | **Mean (± SD)** | **Mean (± SD)** | |
| Decayed (D) | 5.38 (± 6.42) | 6.30 (± 7.25) | 0.076 |
| Missing (M) | 5.89 (± 9.93) | 11.68 (±16.94) | <0.001* |
| Filled (F) | 7.44 (± 11.89) | 10.81 (± 14.05) | 0.272 |
| Total | 18.71 (± 18.34) | (± 25.43) | <0.001* |
| **Plaque Assessment** | **Firm Processing Likers** | **Soft Processing Likers** | **P-Value** |
| **(Plaque index of Silness and Loe)#** | **Mean ± SD** | **Mean ± SD** | |
| Upper Right Quadrant | 0.58 (± 0.64) | 0.59 (± 0.63) | 0.910 |
| Upper Left Quadrant | 0.56 (± 0.64) | 0.55 (± 0.65) | 0.836 |
| Upper Jaw | 0.57 (± 0.63) | 0.57 ± 0.63 | 0.978 |
| Lower Right Quadrant | 0.64 (± 0.63) | 0.64 (± 0.65) | 0.902 |
| Lower Left Quadrant | 0.64 (± 0.62) | 0.64 (± 0.62) | 0.986 |
| Lower Jaw | 0.64 (± 0.62) | 0.64 (± 0.63) | 0.968 |
| Universal | 0.61 (± 0.60) | 0.61 (± 0.60) | 0.964 |

Caries prevalence as scored by the DMFS index and plaque level as assessed by the plaque index of Silness and Loe among firm processing and soft processing likers.

* Adjusted for age, sex, BMI, medical problems, medications, dental visits, brushing and flossing frequency, consumption frequency of sugary drinks, candies, snacks, and sugary baked items, smoking status, participant's education level, parents' education level, income level and the plaque index.

# Plaque levels were assessed per quadrant, jaw, and mouth (universal). Adjusted for age, sex, dental visits, brushing and flossing frequency, consumption frequency of sugary drinks, candies, snacks, and sugary baked items, smoking status, participant's education level, parents' education level, and caries experience.

example, Chaffee et al. [14] and Norton et al. [58] found that the preferred oral processing behavior (mouth behavior) was not influenced by age. On the contrary, there was some disagreement in the literature regarding the relationship between the preferred oral processing behavior and the individual's sex. Similar to our results, Wilson et al. [59] and Zhou et al. [60] demonstrated that the preferred oral processing behavior was not affected by an individual's sex. However, one study reported that more females were typed as crunchers compared to males [14]. The small sample size of crunchers (n = 36) in this study may have led to such a finding.

The third key finding was the existence of a possible relationship between the individual's preferred oral processing behavior and dental caries experience. However, the direction and the nature of such a relationship remain unclear. The preferred oral processing behavior may affect and/ or be affected by dental caries experience. For example, SPL had a higher mean DMFS score (total DMFS) compared to FPL. This indicated higher caries experience among SPL compared to their counterparts. Jeltema et al. and others reported that SPL (smooshers and suckers) exhibited longer eating times compared to FPL [8, 15]. Longer eating time and longer oral retention times are associated with greater caries experience [16, 19, 40]. In these situations, teeth will be exposed to acids for a longer duration. Consequently, the teeth will be subjected to longer periods of demineralization and shorter periods of remineralization [17]. Nevertheless, there are a few caveats to such interpretation. Although the mean DMFS score varied significantly between the different oral processing behaviors, no difference in the mean decayed "D" or filled "F" components of the DMFS was observed between the groups. The former observation could be attributed to the underestimation of dental caries as initial lesions were not recorded and radiographs were not taken. The latter observation was probably related to both groups having similar patterns to using available dental services [61]. Jordan is perceived as a country with limited resources and facing economic challenges [62]. In poor societies, extraction is sought more frequently than restorative treatment [63–65] which likely played a role in the negligible difference observed in the mean score of the "F" component. Therefore, in this study SPL had a greater number of teeth that possibly needed extraction due to caries compared to FPL. This may have contributed to the significant difference in the mean score of the missing "M" component of the DMFS between the groups.

Although the preferred oral processing behavior may affect the dental caries experience, it may be viewed as an outcome of an individual's caries experience. For example, extractions done due to dental caries may have affected the preferred oral processing behavior of an individual. Having a greater number of missing surfaces/teeth due to dental caries might direct an individual's preference into using the palate and tongue rather than teeth during oral processing. Evidence indicates that dental status including the number of missing teeth can affect mastication and oral processing [54–56, 66]. Previous work by Jeltema et al. and others examined the preferred oral processing behavior (mouth behavior groups) in dentate individuals. However, none comprehensively examined the dental status [8, 51, 59] or provided information [12, 67] about the dental health of study participants.

In addition to assessing dental caries experience, dental plaque levels were evaluated among the different oral processing behaviors. The dental plaque (biofilm) is known for its role in the development of dental caries and periodontal disease [68, 69]. Despite being controversial [70], previous evidence indicated that certain food textures such as hard food (including apples) were found to reduce dental plaque accumulation levels [71]. Since variations in food texture preference were previously reported between the oral processing (mouth behavior) groups [7, 8, 11], it was of interest in this project to determine dental plaque levels among the different oral processing behaviors. Although plaque accumulation levels were similar between FPL and SPL, a significant difference in dental caries experience was demonstrated. This was

in agreement with previous reports in which dental plaque accumulation levels demonstrated a positive but weak relationship with the dental caries experience [72–74]. The findings in this study at least partially indicate that oral processing behavior might affect an individual's caries experience by means other than the dental plaque levels. These could be related to variations in biofilm structure and composition [69, 75, 76], or other salivary parameters such as the salivary bacterial viability [77], and salivary flow rate [52]. Indeed, preliminary data indicates a higher salivary flow rate among crunchers and suckers [52]. Although differences in salivary flow rate were observed, the sample size was small (crunchers n = 41, suckers n = 4) [52].

Besides these suggested means, a recent study demonstrated ethnic variation in sweet taste perception among the different oral processing behaviors [13]. In the two groups of the study, Danish and Chinese participants were compared in terms of sweet taste perception. For the Danish group, SPL demonstrated lower sweet taste perception compared to FPL. They provided a lower rating for the sweetness of different yogurt samples when compared to FPL [13]. The opposite was observed in the Chinese study group [13]. Although the relationship between sweet taste perception and oral processing behaviors is still unclear, further research is warranted as evidence suggests that lower sweet taste perception is associated with greater sucrose intake and increased caries experience [78–80]. The fact that the findings of our study did not demonstrate differences in the consumption of sugary drinks does not contradict the previous evidence. The lack of such difference in sugary drinks consumption between FPL and SPL in our sample could be attributed to the method used to collect the frequency of sugary drinks intake (short food frequency questionnaire) [81]. A more comprehensive way of reporting food intake may reveal such differences between the different oral processing behaviors [82, 83].

Although this study provided an insight into a new relationship, there were some limitations to be recognized. The first limitation was related to dental caries assessment. Dental caries was assessed without taking bitewing radiographs. This may have led to underestimation of dental caries experience in both groups. Moreover, the DMFS index was used to assess dental caries experience. The index is easy to apply and is not time-consuming [84, 85]. However, according to the WHO guidelines initial enamel lesions were not recorded [45, 84]. This may have caused an underestimation of dental caries. Conversely, an overestimation of the "M" component may have been caused by recall bias and other biases pertinent to the calculation of the "M" component in individuals ≥ 30 years [44, 86]. While the score of the "M" component depends on losses due to dental caries, other reasons for tooth loss such as trauma, periodontal disease, and orthodontic reasons, are included in the calculations of the "M" component in this age group [45, 86]. To overcome such problems, future studies are encouraged to examine this potential relationship in dentate individuals. Nevertheless, using the DMFS index with no bitewing radiographs was useful for exploring such a potential relationship similar to other studies investigating new associations with dental caries [87–89].

The second limitation was related to the complex etiology of dental caries. Even though this study adjusted for many confounding factors, the multifactorial nature of the caries process makes it unfeasible to control for all confounders in a single study [87, 90–93]. For instance, despite collecting information about the participant's oral hygiene practices, certain details related to the use of fluoride in these practices were not acquired. The anticaries effect of fluoride is influenced by the amount and concentration of fluoride toothpaste, and the rinsing regimen [94–96]. The fact that fluoride use was not accounted for in this study represents a main limitation that may have led to an unwanted bias. However, previous reports have shown that the use of fluoridated toothpaste among Jordanians varied between (∼62%-92%) [97, 98]. Moreover, the pattern of seeking dental care was evaluated, but no data was gathered about the reasons for seeking dental care and the type of treatment received. Evidence suggests that the number of missing teeth may vary depending on the reason for seeking dental care (having

symptoms versus not having symptoms) [99, 100]. While some confounders were partly adjusted for in this study, other confounders were not. For example, genetic variants associated with dental caries [101, 102], salivary flow rate, salivary composition, and buffer capacity [103–106]. The lack of recording these salivary parameters may have been a possible source of bias and a limitation that needs further attention when interpreting the findings of this study.

The third limitation was related to the POPB questionnaire which was adapted from Cattaneo et al. Identifying the subcategories (crunchers, chewers, suckers, smooshers) of each oral processing behavior (FPL, SPL) was unfeasible using this questionnaire. Cattaneo et al. [12] attributed this to the limited sample size in their study. Our sample size was larger ($\sim$ 3 times), yet the cluster analysis did not identify the mouth behavior groups recognized by Jeltema et al. [8, 11]. This could be attributed to differences in the design of the questionnaire [12]. Moreover, It was noticed that the discrimination ability of a few questions (Q12, Q14, Q15, Q17, Q19, Q20) was weak. This may have limited the ability of the questionnaire to identify the 4 mouth behavior groups. For future studies, we suggest using the JBMB typing tool ® if researchers are interested in the subcategories of the preferred oral processing behaviors.

While the study had these limitations, points of strength in this work consisted of the large sample size and the adjustment for a large group of confounding variables. These included the amount of dental plaque [107], age [65, 108], sex [109–111], BMI [108], having medical problems or taking medications [112, 113], frequency of dental visits [114–116], frequency of using oral hygiene aids (tooth brushing and dental flossing) [114, 116], frequency of consuming snacks, high-sugar drinks, candies, and sweet baked items [114, 116], smoking [117], and the socioeconomic status (SES) as measured by income and education levels. Furthermore, a single trained and calibrated dentist examined all study subjects. Moreover, the questionnaire was filled out in person where a trained research assistant was available to provide clarifications when needed. Finally, third molars were excluded from caries and plaque assessments. Dental caries is the most frequent lesion in third molars and is responsible for 15% of third molar extractions [118–120].

In conclusion, an individual's preferred oral processing behavior could potentially be linked to their caries experience. Although the current findings don't support a causal association, they provide a new insight into a possible relationship. A relationship in which oral processing behavior can affect and/or be affected by an individual's caries experience. Future longitudinal studies are needed to confirm this relationship and to understand its nature. Moreover, it is necessary to assess the generalizability of our findings to other ethnic/racial groups. It would be crucial to investigate this potential relationship among the different mouth behavior groups in dentate individuals using the JBMB tool®. Such knowledge would be of relevance for oral health care professionals when offering treatment and prevention. Furthermore, it has a potential impact on the food industry [121]. An impact manifested by designing personalized food products with anticariogenic potential that fulfill the needs of the different oral processing behavior groups.

## Supporting information

**S1 File. Answer key for POPB questionnaire.**
(PDF)

**S2 File. POPB questionnaire.**
(PDF)

**S3 File. Dataset.**
(XLSX)

## Acknowledgments

The authors wish to acknowledge the following team of research assistants for their tremendous efforts during the data collection phase of the study: Nabilah Quadier, Omar Alrawaqah, Reda Elboudali, Saba Al-Hajjeh, Shahd Al-alem, Yousef Trad. We would like to acknowledge as well the help we received from Dr. Aziz Jaber (associate professor of linguistics at Yarmouk University), Prof. Azmi Darwazeh, and Dr. Mohammad Atieh during the translation process of the POPB questionnaire.

## Author Contributions

**Conceptualization:** Melanie F. Alazzam.

**Formal analysis:** Yousef S. Khader.

**Investigation:** Issam B. Rasheed, Dalal A. Shamiyah.

**Methodology:** Melanie F. Alazzam, Issam B. Rasheed, Suhad H. Aljundi, Reem S. Abdelhafez.

**Project administration:** Melanie F. Alazzam.

**Supervision:** Melanie F. Alazzam, Issam B. Rasheed, Suhad H. Aljundi, Reem S. Abdelhafez.

**Writing – original draft:** Melanie F. Alazzam.

**Writing – review & editing:** Issam B. Rasheed, Suhad H. Aljundi, Dalal A. Shamiyah, Yousef S. Khader, Reem S. Abdelhafez, Mohammad S. Alrashdan.

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
