## [Decision Letter · Decision Letter 0]

26 Dec 2023

PONE-D-23-38122Oral processing behavior and dental caries; a new relationshipPLOS ONE

Dear Dr. Alazzam,

Thank you for submitting your manuscript to PLOS ONE. After careful consideration, we feel that it has merit but does not fully meet PLOS ONE’s publication criteria as it currently stands. Therefore, we invite you to submit a revised version of the manuscript that addresses the points raised during the review process.

We look forward to receiving your revised manuscript.

Kind regards,

Hadi Ghasemi

Academic Editor

PLOS ONE

Journal Requirements:

Changes in Distance Running Mechanics Due to Systematic Variations in Running Style - https://doi.org/10.1123/ijsb.7.1.76

Effectiveness of Lower-Cost Strategies for Running Gait Retraining: A Systematic Review - https://doi.org/10.3390/app13031376

(among others)

In your revision ensure you cite all your sources (including your own works), and quote or rephrase any duplicated text outside the methods section. Further consideration is dependent on these concerns being addressed.

3. Please include your tables as part of your main manuscript and remove the individual files. Please note that supplementary tables (should remain/ be uploaded) as separate ""supporting information"" files

Reviewers' comments:

Reviewer's Responses to Questions

**Comments to the Author**

1. Is the manuscript technically sound, and do the data support the conclusions?

Reviewer #1: Yes

Reviewer #2: Partly

Reviewer #3: Yes

2. Has the statistical analysis been performed appropriately and rigorously? 

Reviewer #1: Yes

Reviewer #2: Yes

Reviewer #3: I Don't Know

3. Have the authors made all data underlying the findings in their manuscript fully available?

Reviewer #1: Yes

Reviewer #2: No

Reviewer #3: Yes

4. Is the manuscript presented in an intelligible fashion and written in standard English?

Reviewer #1: Yes

Reviewer #2: Yes

Reviewer #3: Yes

5. Review Comments to the Author

Reviewer #1: Dear Authors

1-The research topic is well-chosen, interesting and important in dental caries prevention policy and behavioral management in your context.

2. The methodology is well-designed and appropriate for the research questions followed by an acceptable data analysis

3. You have effectively integrated previous research and contextualized their work.

4. The manuscript is well-structured, which increases the readability of the article.

5. Please include the POPB questionnaire.

6. Using a translated questionnaire for study requires standardization of the translated version (translation, back translation, expert opnion...). Please explain more if you have standardized the question.

Reviewer #2: Comments regarding the manusript entitled "Oral processing behaviour and dental caries; a new relationship".

I think the manuscript is very interesting, and it provides new insights regarding the caries disease. However, there are some major concerns.

Inital lesions were not recorded, and radiographic imaging was not used, why? As the authors must know, without radiographs there will be problems with false negative outcomes. I know this is discussed in the discussion but this is truly a major issue.

DMFT was significantly higher in one of the groups, however, when looking at the individual aspects, only the M = missing was significantly higher. Is it a correct conclusion to draw that dental caries has a correlation with eating behaviours in this study? For me, it is rather oral health and not specifically dental caries that can be discussed since decayed and filled were not significantly higher in one of the groups. The limitations of the DMFT-index is that the M = missing part is always unknown regarding the aethiology.

In the manuscript, the aspect of seeking dental care is discussed, and how this affects the results. The question of seeking dental care, and specifically asking about why and what kind of care (only emergency, regular check-ups, filling therapy) should have been included in the questionnaire.

The largest limitation in this manuscript is that the fluoride aspect has not been accounted for. Toothbrushing was included in the questionnaire, but not fluoride concentrations in toothpaste, toothpaste behaviours (such as amounts of fluoride toothpaste used, rinsing with water during or after brushing), or usage of additional fluoride products. The usage of fluoride in the treatment and prevention of dental caries is of great importance, and probably affects the results in this study. This is not even discussed in the manuscript, why?

Additionally, saliva secretion rates, and buffer capacity has not been included either which is a limitation since this affects the caries risk.

Regarding supplementary information, I cannot see the pictures used in the questionnaire about food preferences, and I cannot find the questionnaire used for collecting the general information.

Reviewer #3: thank you for your work, I have a few comments to address prior to publication:

-Would it have been meaningful to include patients with minimal missing teeth or possible none to eliminate this confounding factor?

-with the information you have collected would it not have been more meaningful to perform caries risk assessment? I encourage you to present it in caries risk format if you are able to. It would simplify data presentation and make comparisons with literature easier since most studies are reporting in that way.

what is the importance of collecting plaque indices since its relationship with dental caries has proven to be weak

6. PLOS authors have the option to publish the peer review history of their article (what does this mean?). If published, this will include your full peer review and any attached files.

Reviewer #1: No

Reviewer #2: No

Reviewer #3: No

---

## [Author Response · Author response to Decision Letter 0]

23 Feb 2024

Dear Dr. Ghasemi,

We would like to thank our reviewers for taking the time and effort to review this work. Their feedback is highly appreciated, and it has truly contributed to improving the quality of our manuscript. In what follows, we addressed each comment raised. In general, we made some modifications to the tone of the manuscript to ensure that our discussion and conclusions accurately reflect the findings of the study, while taking into consideration the limitations of our research. 

Kindly note that our response is uploaded in a separate file labelled as "Response to Reviewers"

Looking forward to hearing from you!

Sincerely, 

Melanie Alazzam

---

## [Decision Letter · Decision Letter 1]

8 Mar 2024

PONE-D-23-38122R1Oral processing behavior and dental caries; an insight into a new relationshipPLOS ONE

Dear Dr. Alazzam,

Thank you for submitting your manuscript to PLOS ONE. After careful consideration, we feel that it has merit but does not fully meet PLOS ONE’s publication criteria as it currently stands. Therefore, we invite you to submit a revised version of the manuscript that addresses the points raised during the review process.

In my perspective, your research warrants attention and publication due to the originality of its concept. In the last paragraph of the M&M (lines 255-256), however, I noticed that you mentioned about a GLM analysis but I did not find any related table or text in the result section in this regard. As you have a lot of confounding variables in your study, it is essential to perform a multivariable analysis like a binary logistic regression. Please, add a table and explain it in the text of the result section.

Fortunately, my viewpoint about the merits of your study is supported by three referees. Simultaneously, one of the referees has provided recommendations that ought to be taken into account and implemented, if feasible. Please consider and apply the comments of that reviewer as far as possible. Just, do not consider the following comment since it is completely inappropriate and not related:

Few paragraphs should be incorporated to introduction about the details of other oral diseases and their diagnosis and treatments. suggestive reference:Aldelaimi TN, Abood SW, Khalil AA. The Evaluation of Impacted Third Molars Using Panoramic Radiograph. Al-Anbar Medical Journal, 2010, 8(1): 26-33Aldelaimi, Tahrir N., and Afrah A. Khalil. "Clinical application of diode laser (980 nm) in maxillofacial surgical procedures." Journal of Craniofacial Surgery 26.4 (2015): 1220-1223Afrah A Aldelaimi, Hamid H Enezei, Tahrir N Aldelaimi, Khalil Abdulla Mohammed, Tumors of Craniofacial Region in Iraq (Clinicopathological Study), J Res Med Dent Sci, 2021, 9 (1): 66-71

We look forward to receiving your revised manuscript.

Kind regards,

Hadi Ghasemi

Academic Editor

PLOS ONE

Journal Requirements:

Reviewers' comments:

Reviewer's Responses to Questions

**Comments to the Author**

1. If the authors have adequately addressed your comments raised in a previous round of review and you feel that this manuscript is now acceptable for publication, you may indicate that here to bypass the “Comments to the Author” section, enter your conflict of interest statement in the “Confidential to Editor” section, and submit your "Accept" recommendation.

Reviewer #4: (No Response)

Reviewer #5: All comments have been addressed

Reviewer #6: (No Response)

2. Is the manuscript technically sound, and do the data support the conclusions?

Reviewer #4: Partly

Reviewer #5: Yes

Reviewer #6: Yes

3. Has the statistical analysis been performed appropriately and rigorously? 

Reviewer #4: Yes

Reviewer #5: Yes

Reviewer #6: I Don't Know

4. Have the authors made all data underlying the findings in their manuscript fully available?

Reviewer #4: Yes

Reviewer #5: Yes

Reviewer #6: Yes

5. Is the manuscript presented in an intelligible fashion and written in standard English?

Reviewer #4: Yes

Reviewer #5: Yes

Reviewer #6: Yes

6. Review Comments to the Author

Reviewer #4: Dear authors

Thanks for sharing your work with us, the following are the suggested changes and comments that should be kept in consideration:

Manuscript title: Oral processing behavior and dental caries; a new relationship.

The comments

• Both the quality and data presentation of this manuscript are acceptable and of great importance to clinicians as well as patients.

• The manuscript expands our knowledge about oral processing behavior and oral diseases including dental caries

• The title should be revised and reduced its characters ( precise & and informative)

• The abstract should reflect the content of the article and must be with range of 250-300 words.

• Three to ten keywords representing the main content of the article

• Few paragraphs should be incorporated to introduction about the details of other oral diseases and their diagnosis and treatments. suggestive reference:

Aldelaimi TN, Abood SW, Khalil AA. The Evaluation of Impacted Third Molars Using Panoramic Radiograph. Al-Anbar Medical Journal, 2010, 8(1): 26-33

Aldelaimi, Tahrir N., and Afrah A. Khalil. "Clinical application of diode laser (980 nm) in maxillofacial surgical procedures." Journal of Craniofacial Surgery 26.4 (2015): 1220-1223

Afrah A Aldelaimi, Hamid H Enezei, Tahrir N Aldelaimi, Khalil Abdulla Mohammed, Tumors of Craniofacial Region in Iraq (Clinicopathological Study), J Res Med Dent Sci, 2021, 9 (1): 66-71

• Although the results are impressive and interesting but need to be revised to be more informative by precised presentation of your finding.

• The statements in discussion are acceptable but more paragraphs about the justification of your findings and comparison with other recent relevant studies.

• Up to date references should be kept in your reference list and the old should be omitted.

Good Luck

Reviewer #5: Dear Authors

Thank you for considering the Plot One journal for your current submission.

the recommendation was send to the editor with minor changing suggestions

good luck

Reviewer #6: Dear authors,

Thank you for your great work, very interesting topic. I believe this study would add a great value to the field of socio-behavioral determinants of dental/oral health.

best of luck

7. PLOS authors have the option to publish the peer review history of their article (what does this mean?). If published, this will include your full peer review and any attached files.

Reviewer #4: No

Reviewer #5: No

Reviewer #6: **Yes: **Farrah Alnajjar

---

## [Author Response · Author response to Decision Letter 1]

16 Apr 2024

Dear Dr. Ghasemi,

We would like to thank you and the reviewers for your interest in our work. We value your feedback and your role in ensuring the quality and integrity of any original research to be published at PLOS One. We would like to thank reviewer 4 for the recommendations which were helpful in further improving our manuscript. In what follows, we addressed each comment raised. In general, we made a few amendments to the methods, results, discussion, and references sections. Moreover, we corrected a few typos, grammatical, punctuation marks, and spelling errors that were discovered while reviewing the manuscript. 

Looking forward to hearing from you!

Sincerely, 

Melanie Alazzam

---

## [Decision Letter · Decision Letter 2]

23 Apr 2024

PONE-D-23-38122R2Oral processing behavior and dental caries; an insight into a new relationshipPLOS ONE

Dear Dr. Alazzam,

Thank you for submitting your manuscript to PLOS ONE. After careful consideration, we feel that it has merit but still does not fully meet PLOS ONE’s publication criteria as it currently stands. Therefore, we invite you to submit a revised version of the manuscript that addresses the points raised during the review process.

We look forward to receiving your revised manuscript.

Kind regards,

Hadi Ghasemi

Academic Editor

PLOS ONE

Journal Requirements:

Reviewers' comments:

Reviewer's Responses to Questions

**Comments to the Author**

1. If the authors have adequately addressed your comments raised in a previous round of review and you feel that this manuscript is now acceptable for publication, you may indicate that here to bypass the “Comments to the Author” section, enter your conflict of interest statement in the “Confidential to Editor” section, and submit your "Accept" recommendation.

Reviewer #2: (No Response)

Reviewer #3: (No Response)

Reviewer #4: All comments have been addressed

2. Is the manuscript technically sound, and do the data support the conclusions?

Reviewer #2: Partly

Reviewer #3: Partly

Reviewer #4: Partly

3. Has the statistical analysis been performed appropriately and rigorously? 

Reviewer #2: Yes

Reviewer #3: I Don't Know

Reviewer #4: Yes

4. Have the authors made all data underlying the findings in their manuscript fully available?

Reviewer #2: Yes

Reviewer #3: Yes

Reviewer #4: No

5. Is the manuscript presented in an intelligible fashion and written in standard English?

Reviewer #2: Yes

Reviewer #3: Yes

Reviewer #4: Yes

6. Review Comments to the Author

**Reviewer #2: **Comments regarding: "Oral processing behaviour and dental caries; a new relationship".

The manuscript has now been revised and is much improved.

I still think the manuscript is very interesting, and it provides new insights regarding oral health related to oral processing behaviours.

I still consider the largest limitation in this manuscript to be that the fluoride aspect has not been accounted for in the method. However, this has now been addressed in the discussion. I am still concerned that excluding this factor might have provided a large bias in the study.

Additionally, saliva secretion rates, and buffer capacity has not been included either which is a limitation, since this greatly affects the caries risk. I have commented on this previously. Saliva flow rates related to oral processing behaviours his is briefly discussed in the discussion, aknowledging that there might be differences between the studied groups, but authors should recognize that the neglect of measuring this factor in current study is as a limitation.

In the discussion, (line 446 and forwards) reference 13 is used. This paragraph is confusing. The study by Liu et al 2021 does not support the statement that SPL demonstrated generally lower sweet taste perceptions compared to FPL. In the two groups of the study, Danish and Chinese test persons were compared. For Chinese individuals, the FPL:s demonstrated lower sweet taste perception compared to SPL:s, but the results were the opposite for Danish test persons. The relationship between oral processing behaviour and sweet taste perception is therefore, at least based on the referred study, not clear. I think this paragraph must be rewritten.

**Reviewer #3:** It is unclear where the modification have taken place. The authors response does not detail that. Kindly respond to each question in the authors response letter stating where the modifications happened in the mauscript.

**Reviewer #4:** • Still more recent studies should be mentioned and more paragraphs should be added to the introduction and discussion with other relevant studies

7. PLOS authors have the option to publish the peer review history of their article (what does this mean?). If published, this will include your full peer review and any attached files.

Reviewer #2: No

Reviewer #3: No

Reviewer #4: **Yes: **Prof. Dr. Tahrir N. Aldelaimi

---

## [Author Response · Author response to Decision Letter 2]

30 Apr 2024

Melanie Alazzam

Corresponding Author

Assistant Professor

Department of Oral Medicine and Oral Surgery

Faculty of Dentistry 

Jordan University of Science and Technology

P.O. Box: 3030, Irbid 22110, Jordan

Telephone: (+962)2-7201000 Ext: 23558

Email address: mfalazam@just.edu.jo

Dr. Hadi Ghasemi

Academic Editor

PLOS ONE

4.30.2024

Dear Dr. Ghasemi, 

We hope that this version meets our reviewers’ expectations and addresses their comments. Kindly note that the study is providing an insight into a new potential relationship between oral processing behavior and dental caries. A relationship in which oral processing behavior can affect and/or be affected by an individual’s caries experience. The findings of our study represent a potential area for future research in the field of food oral processing and /or the dental caries process. We kindly ask our reviewers to read the manuscript in light of the study aims, taking into consideration the sample size and the number of confounding factors that were adjusted for in our study. Understanding the multifactorial nature of dental caries guided us into a more humble conclusion and further encouraged us to further investigate this relationship taking into consideration other important confounders that our reviewers have referred to. 

In this revision, we made minor changes to elaborate on a few limitations in our study. We have also rewritten a paragraph in the discussion section. 

Thank you for taking the time and effort to help us improve this interesting work.

Looking forward to hearing from you!

Sincerely, 

Melanie Alazzam

---

## [Decision Letter · Decision Letter 3]

17 May 2024

PONE-D-23-38122R3Oral processing behavior and dental caries; an insight into a new relationshipPLOS ONE

Dear Dr. Alazzam,

Thank you for submitting your manuscript to PLOS ONE. After careful consideration, we feel that it has merit but does not fully meet PLOS ONE’s publication criteria as it currently stands. Therefore, we invite you to submit a revised version of the manuscript that addresses the points raised during the review process.

We look forward to receiving your revised manuscript.

Kind regards,

Hadi Ghasemi

Academic Editor

PLOS ONE

Journal Requirements:

Reviewers' comments:

Reviewer's Responses to Questions

**Comments to the Author**

1. If the authors have adequately addressed your comments raised in a previous round of review and you feel that this manuscript is now acceptable for publication, you may indicate that here to bypass the “Comments to the Author” section, enter your conflict of interest statement in the “Confidential to Editor” section, and submit your "Accept" recommendation.

Reviewer #2: All comments have been addressed

Reviewer #3: (No Response)

2. Is the manuscript technically sound, and do the data support the conclusions?

Reviewer #2: Yes

Reviewer #3: Partly

3. Has the statistical analysis been performed appropriately and rigorously? 

Reviewer #2: Yes

Reviewer #3: I Don't Know

4. Have the authors made all data underlying the findings in their manuscript fully available?

Reviewer #2: Yes

Reviewer #3: Yes

5. Is the manuscript presented in an intelligible fashion and written in standard English?

Reviewer #2: Yes

Reviewer #3: Yes

6. Review Comments to the Author

Reviewer #2: I have again reviewed the manuscript. As a whole, I think the manuscript now is suitable for publication. However, I have a few comments.

In the discussion, there is a minor language error. Line 422: “Despite being controversial [68], hard food (including apples) was found to reduce the accumulation levels of dental plaque [69]”. This statement is referring to a study, thus this should be clearly stated : “Despite being controversial [68], a study has found that…".

In contrary to the authors, I am not surprised by the findings that there was no difference in dental caries prevalence related to plaque levels. Several previous studies have shown that in general, there are only weak correlations between dental caries and dental plaque levels, except from root caries where a stronger correlation can be observed.

On line 475, a statement has been included to clarify the bias of not accounting for fluoride in the study. However, I think this statement should be rephrased (referring to the authors themselves in the text as “we” is not proper academic language).

As a side note, I must say that I am surprised that the authors have not addressed and responded to the reviewers’ questions one by one, as is expected when resubmitting a revised manuscript.

Reviewer #3: Kindly respond in detail to original comments while making the change clear in the authors response. for example: materials in methods, line 2: change is the so and so

7. PLOS authors have the option to publish the peer review history of their article (what does this mean?). If published, this will include your full peer review and any attached files.

Reviewer #2: No

Reviewer #3: No

---

## [Author Response · Author response to Decision Letter 3]

28 May 2024

Dear Dr. Ghasemi, 

We thank you and the reviewers for your continued support and interest in our manuscript. We would like to confirm that in all 4 revisions of this manuscript, we provided a “response to reviewers” file. The file in each of the 4 revisions contained a cover letter and a table that included 5 columns (Reviewer number, Comment number, Reviewer’s comment, Authors’ reply, Author’s action). The cover letter and the table were in landscape style. We are concerned that for a technical reason, the “response to reviewers” file did not appear for the reviewers last time. We remember encountering a note at PLOS One editorial system which indicated that there were some technical issues with the website. We deeply appreciate the effort our reviewers put into this review process and believe that not addressing each comment is disrespectful to our colleague researchers. Therefore, we were/are always very keen to address each and every comment received. 

Below, we provide our response to the last revision we received. In addition, we provide our response to the comments that we expect were provided by reviewers “2” and “3” in previous revisions as well. 

Please note that the comments were organized in a table as we have just described above. We highlighted the comments provided by reviewer “2” in light blue and the comments provided by reviewer “3” in yellow. 

We thank you for your understanding and we look forward to hearing from you!

Sincerely, 

Melanie Alazzam

---

## [Decision Letter · Decision Letter 4]

11 Jun 2024

Oral processing behavior and dental caries; an insight into a new relationship

PONE-D-23-38122R4

Dear Dr. Melanie F. Alazzam,

We’re pleased to inform you that your manuscript has been judged scientifically suitable for publication and will be formally accepted for publication once it meets all outstanding technical requirements.

Kind regards,

Hadi Ghasemi

Academic Editor

PLOS ONE

Additional Editor Comments (optional):

Reviewers' comments:

Reviewer's Responses to Questions

**Comments to the Author**

1. If the authors have adequately addressed your comments raised in a previous round of review and you feel that this manuscript is now acceptable for publication, you may indicate that here to bypass the “Comments to the Author” section, enter your conflict of interest statement in the “Confidential to Editor” section, and submit your "Accept" recommendation.

Reviewer #3: All comments have been addressed

2. Is the manuscript technically sound, and do the data support the conclusions?

Reviewer #3: Yes

3. Has the statistical analysis been performed appropriately and rigorously? 

Reviewer #3: I Don't Know

4. Have the authors made all data underlying the findings in their manuscript fully available?

Reviewer #3: Yes

5. Is the manuscript presented in an intelligible fashion and written in standard English?

Reviewer #3: Yes

6. Review Comments to the Author

Reviewer #3: Thanks for addressing my comments. Hopefully, feedback can help design a more improved follow up study.

7. PLOS authors have the option to publish the peer review history of their article (what does this mean?). If published, this will include your full peer review and any attached files.

Reviewer #3: No

---

## [Editor Report · Acceptance letter]

24 Jun 2024

PONE-D-23-38122R4 

PLOS ONE

Dear Dr. Alazzam, 

I'm pleased to inform you that your manuscript has been deemed suitable for publication in PLOS ONE. Congratulations! Your manuscript is now being handed over to our production team.

Kind regards, 

on behalf of

Dr. Hadi Ghasemi 

Academic Editor

PLOS ONE